# Empirical Frequentist Coverage of Deep Learning Uncertainty Quantification Procedures

**DOI:** 10.3390/e23121608

**Published:** 2021-11-30

**Authors:** Benjamin Kompa, Jasper Snoek, Andrew L. Beam

**Affiliations:** 1Department of Biomedical Informatics, Harvard Medical School, Boston, MA 02115, USA; benjamin_kompa@hms.harvard.edu; 2Google Research, Cambridge, MA 02142, USA; jsnoek@google.com; 3Department of Epidemiology, Harvard School of Public Health, Boston, MA 02115, USA

**Keywords:** uncertainty quantification, coverage, Bayesian methods, dataset shift

## Abstract

Uncertainty quantification for complex deep learning models is increasingly important as these techniques see growing use in high-stakes, real-world settings. Currently, the quality of a model’s uncertainty is evaluated using point-prediction metrics, such as the negative log-likelihood (NLL), expected calibration error (ECE) or the Brier score on held-out data. Marginal coverage of prediction intervals or sets, a well-known concept in the statistical literature, is an intuitive alternative to these metrics but has yet to be systematically studied for many popular uncertainty quantification techniques for deep learning models. With marginal coverage and the complementary notion of the width of a prediction interval, downstream users of deployed machine learning models can better understand uncertainty quantification both on a global dataset level and on a per-sample basis. In this study, we provide the first large-scale evaluation of the empirical frequentist coverage properties of well-known uncertainty quantification techniques on a suite of regression and classification tasks. We find that, in general, some methods do achieve desirable coverage properties on *in distribution* samples, but that coverage is not maintained on out-of-distribution data. Our results demonstrate the failings of current uncertainty quantification techniques as dataset shift increases and reinforce coverage as an important metric in developing models for real-world applications.

## 1. Introduction

Predictive models based on deep learning have seen a dramatic improvement in recent years [1], which has led to widespread adoption in many areas. For critical, high-stakes domains, such as medicine or self-driving cars, it is imperative that mechanisms are in place to ensure safe and reliable operation. Crucial to the notion of safe and reliable deep learning is the effective quantification and communication of predictive uncertainty to potential end-users of a system. In medicine, for instance, understanding predictive uncertainty could lead to better decision-making through improved allocation of hospital resources, detecting dataset shift in deployed algorithms, or helping machine learning models abstain from making a prediction [2]. For medical classification problems involving many possible labels (i.e., creating a differential diagnosis), methods that provide a set of possible diagnoses when uncertain are natural to consider and align more closely with the differential diagnosis procedure used by physicians. The prediction sets and intervals we propose in this work are an intuitive way to quantify uncertainty in machine learning models and provide interpretable metrics for downstream, nontechnical users.

Commonly used approaches to quantify uncertainty in deep learning generally fall into two broad categories: ensembles and approximate Bayesian methods. Deep ensembles [3] aggregate information from multiple individual models to provide a measure of uncertainty that reflects the ensembles’ agreement about a given data point. Bayesian methods offer direct access to predictive uncertainty through the posterior predictive distribution, which combines prior knowledge with the observed data. Although conceptually elegant, calculating exact posteriors of even simple neural models is computationally intractable [4,5], and many approximations have been developed [6,7,8,9,10,11,12]. Though approximate Bayesian methods scale to modern sized data and models, recent work has questioned the quality of the uncertainty provided by these approximations [4,13,14].

Previous work assessing the quality of uncertainty estimates has focused on calibration metrics and scoring rules, such as the negative log-likelihood (NLL), expected calibration error (ECE), and Brier score. Here we provide an alternative perspective based on the notion of empirical coverage, a well-established concept in the statistical literature [15] that evaluates the quality of a predictive set or interval instead of a point prediction. Informally, coverage asks the question: If a model produces a predictive uncertainty interval, how often does that interval actually contain the observed value? Ideally, predictions on examples for which a model is uncertain would produce larger intervals and thus be more likely to cover the observed value.

In this work, we focus on marginal coverage *over a dataset* $D^{\prime }$ for the canonical α value of 0.05, i.e., 95% prediction intervals. For a machine learning model that produces a 95% prediction interval $\hat{C_{n}}\left(x_{n}\right)$ based on the training dataset D, we consider what fraction of the points in the dataset $D^{\prime }$ have their true label contained in $\hat{C_{n}}\left(x_{n+1}\right)$ for $x_{n+1}\in D^{\prime }$. To measure the robustness of these intervals, we also consider cases when the generating distributions for D and $D^{\prime }$ are not the same (i.e., dataset shift).

Figure 1 provides a visual depiction of marginal coverage over a dataset for two hypothetical regression models. Throughout this work, we refer to “marginal coverage over a dataset” as “coverage”.

For a machine learning model that produces predictive uncertainty estimates (i.e., approximate Bayesian methods and ensembling), coverage encompasses both the aleatoric and epistemic uncertainties [16] produced by these models. In a regression setting, the predictions from these models can be written as:(1)$$\hat{y}=f\left(x\right)+\epsilon$$
where epistemic uncertainty is captured in the f(x) component, while aleatoric uncertainty is considered in the ϵ term. Since coverage captures how often the predicted interval of $\hat{y}$ contains the true value, it captures the contributions from both types of uncertainty.

A complementary metric to coverage is width, which is the size of the prediction interval or set. In regression problems, we typically measure width in terms of the standard deviation of the true label in the training set. As an example, an uncertainty quantification procedure could produce prediction intervals that have 90% marginal coverage with an average width of two standard deviations. For classification problems, width is simply the average size of a prediction set. Width can provide a relative ranking of different methods, i.e., given two methods with the same level of coverage, we should prefer the method that provides intervals with smaller widths.

**Contributions:** In this study, we investigate the empirical coverage properties of prediction intervals constructed from a catalog of popular uncertainty quantification techniques, such as ensembling, Monte Carlo dropout, Gaussian processes, and stochastic variational inference. We assess the coverage properties of these methods on nine regression tasks and two classification tasks with and without dataset shift. These tasks help us make the following contributions:We introduce coverage and width over a dataset as natural and interpretable metrics for evaluating predictive uncertainty for deep learning models.A comprehensive set of coverage evaluations on a suite of popular uncertainty quantification techniques.An examination of how dataset shift affects these coverage properties.

## 2. Background and Related Work

### 2.1. Frequentist Coverage and Conformal Inference

Given features $x_{i}\in R^{d}$ and a response $y_{i}\in R$ for some dataset $D={\left\{\left(x_{i},y_{i}\right)\right\}}_{i=1}^{n}$, Barber et al. [17] define *distribution-free* marginal coverage in terms of a set ${\hat{C}}_{n}\left(x\right)$ and a level α∈[0,1]. The set ${\hat{C}}_{n}\left(x\right)$ is said to have coverage at the 1−α level if for all distributions *P* such that $\left(x,y\right)\in R^{d}\times R$ and (x,y)∼P, the following inequality holds:(2)$$P\{y_{n+1}\in {\hat{C}}_{n}\left(x_{n+1}\right)\}\geq 1-\alpha$$

For new samples beyond the first *n* samples in the training data, there is a 1−α probability of the true label of the test point being contained in the set ${\hat{C}}_{n}\left(x_{n+1}\right)$. This set can be constructed using a variety of procedures. For example, in the case of simple linear regression, a prediction interval for a new point $x_{n+1}$ can be constructed using a simple, closed-form solution [15].

Marginal coverage is typically considered in the limit of infinite samples. However, here we focus on marginal coverage over a dataset D. We assess, for a given model and test set D, the empirical coverage by assessing whether $y_{n+1}\in {\hat{C}}_{n}\left(x_{n+1}\right)\;\forall x_{n+1}\in D$. Additionally, we consider how marginal coverage changes as there is data distribution shift such that a new dataset $D^{\prime }$ has a different data generating distribution. Despite the lack of infinite samples, this work establishes the motivation of considering coverage in critical, high-risk situations, such as medicine.

An important and often overlooked distinction is that of marginal and conditional coverage. In conditional coverage, one considers:(3)$$P\{y_{n+1}\in {\hat{C}}_{n}\left(x_{n+1}\right)|x_{n+1}=x\}\geq 1-\alpha$$

The probability has been conditioned on specific features. This is potentially a more useful version of coverage to consider because one could make claims for specific instances rather than over the broader distribution *P*. However, it is impossible in general to have conditional coverage guarantees [17].

Conformal inference [18,19] is one statistical framework that can provide marginal coverage under a certain set of assumptions (e.g., exchangeable data) that we do not assume here [20]. In this work, we specifically seek to measure the empirical coverage of the existing approximate Bayesian and alternative uncertainty quantification methods with and without dataset shift. These methods are extremely popular in practice, but nobody has yet considered the empirical coverage of their 95% posteriors. Conformal methods are not part of the approximate Bayesian methods that we set out to analyze in this work. There has been recent work on Bayes-optimal prediction with frequentist coverage control [21] and conformal inference under dataset shift [22,23]. However, adding the conformal framework to approximate Bayesian methods post hoc and measuring their coverage properties could be interesting future work. An additional distinction between our work and the broader conformal inference literature is that we do not aim to provide finite sample coverage guarantees.

Another important point to consider is that while the notion of a confidence interval may seem natural to consider in our analysis, confidence intervals estimate global statistics over repeated trials of data and generally come with guarantees about how often these statistics lie in said intervals. In our study, this is not the case. Although we estimate coverage across many datasets, we are not aiming to estimate an unknown statistic of the data. We would like to understand the empirical coverage properties of machine learning models.

### 2.2. Obtaining Predictive Uncertainty Estimates

Several lines of work focus on improving approximations of the posterior of a Bayesian neural network [6,7,8,9,10,11,12]. Yao et al. [4] provide a comparison of many of these methods and highlight issues with common metrics of comparison, such as test-set log-likelihood and RMSE. Good scores on these metrics often indicate that the model posterior happens to match the test data rather than the true posterior [4]. Maddox et al. [24] developed a technique to sample the approximate posterior from the first moment of stochastic gradient descent iterates. Wenzel et al. [13] demonstrated that despite advances in these approximations, in practice, approximate methods for the Bayesian modeling of deep networks do not perform as well as theory would suggest.

Alternative methods that do not rely on estimating a posterior over the weights of a model can also be used to provide uncertainty estimates. Gal and Ghahramani [16], for instance, demonstrated that Monte Carlo dropout is related to a variational approximation to the Bayesian posterior implied by the dropout procedure. Lakshminarayanan et al. [3] used an ensemble of several neural networks to obtain uncertainty estimates. Guo et al. [25] established that temperature scaling provides well-calibrated predictions on an i.i.d test set. More recently, van Amersfoort et al. [26] showed that the distance from the centroids in an RBF neural network yields high-quality uncertainty estimates. Liu et al. [27] also leveraged the notion of distance (in the form of an approximate Gaussian process covariance function) to obtain uncertainty estimates with their Spectral-normalized Neural Gaussian Processes.

### 2.3. Assessments of Uncertainty Properties under Dataset Shift

Ovadia et al. [14] analyzed the effect of dataset shift on the accuracy and calibration of a variety of deep learning methods. Their large-scale empirical study assessed these methods on standard datasets, such as MNIST, CIFAR-10, ImageNet, and other non-image-based datasets. Additionally, they used translations, rotations, and corruptions of these datasets [28] to quantify performance under dataset shift. They found stochastic variational inference (SVI) to be promising on simpler datasets, such as MNIST and CIFAR-10, but more difficult to train on larger datasets. Deep ensembles had the most robust response to dataset shift.

## 3. Methods

For features $x_{i}\in R^{d}$ and a response $y_{i}\in R$ or $y_{i}\in Z$ (for regression and classification, respectively) for some dataset $D={\left\{\left(x_{i},y_{i}\right)\right\}}_{i=1}^{n}$, we consider the prediction intervals or sets ${\hat{C}}_{n}\left(x\right)$ in regression and classification settings, respectively. Unlike in the definitions of marginal and conditional coverage, we do not assume that (x,y)∼P always holds true. Thus, we consider the marginal coverage on a dataset $D^{\prime }$ for some new test sets that may have undergone dataset shift from the generating distribution of the training set D.

In both the regression and classification settings, we analyzed the coverage properties of prediction intervals and sets of five different approximate Bayesian and non-Bayesian approaches for uncertainty quantification. These include dropout [16,29], ensembles [3], Stochastic Variational Inference [7,8,11,12,30], and last layer approximations of SVI and dropout [31]. Additionally, we considered prediction intervals from linear regression and the 95% credible interval of a Gaussian process with the squared exponential kernel as baselines in regression tasks. For classification, we also considered temperature scaling [25] and the softmax output of vanilla deep networks [28]. For more detail on our modeling choices, see Appendix B.

### 3.1. Regression Methods and Metrics

We evaluated the coverage properties of these methods on nine large real-world regression datasets used as a benchmark in Hernández-Lobato and Adams [6] and later Gal and Ghahramani [16]. We used the training, validation, and testing splits publicly available from Gal and Ghahramani [16] and performed nested cross-validation to find hyperparameters. On the training sets, we did 100 trials of a random search over hyperparameter space of a multi-layer-perceptron architecture with an Adam optimizer [32] and selected hyperparameters based on RMSE on the validation set.

Each approach required slightly different ways to obtain a 95% prediction interval. For an ensemble of neural networks, we trained N=40 vanilla networks and used the 2.5% and 97.5% quantiles as the boundaries of the prediction interval. For dropout and last layer dropout, we made 200 predictions per sample and similarly discarded the top and bottom 2.5% quantiles. For SVI, last layer SVI (LL SVI), and Gaussian processes we had approximate variances available for the posterior, which we used to calculate the prediction interval. We calculated 95% prediction intervals from linear regression using the closed-form solution.

Then we calculated two metrics:**Coverage**: A sample is considered covered if the true label is contained in this 95% prediction interval. We average over all samples in a test set to estimate a method’s marginal coverage on this dataset.**Width**: The width is the average over the test set of the ranges of the 95% prediction intervals.

Coverage measures how often the true label is in the prediction region, while width measures how specific that prediction region is. Ideally, we would have high levels of coverage with low levels of width on in-distribution data. As data becomes increasingly out of distribution, we would like coverage to remain high while width increases to indicate model uncertainty.

### 3.2. Classification Methods and Metrics

Ovadia et al. [14] evaluated model uncertainty on a variety of datasets publicly available. These predictions were made with the five approximate Bayesian methods described above, plus vanilla neural networks, with and without temperature scaling. We focus on the predictions from MNIST, CIFAR-10, CIFAR-10-C, ImageNet, and ImageNet-C datasets. For MNIST, we calculated coverage and width of model prediction intervals on rotated and translated versions of the test set. For CIFAR-10, Ovadia et al. [14] measured model predictions on translated and corrupted versions of the test set from CIFAR-10-C [28] (see Figure 2). For ImageNet, we only considered the coverage and width of prediction sets on the corrupted images of ImageNet-C [28]. Each of these transformations (rotation, translation, or any of the 16 corruptions) has multiple levels of shift. Rotations range from 15 to 180 degrees in 15 degrees increments. Translations shift images every 2 and 4 pixels for MNIST and CIFAR-10, respectively (see Figure 3). Corruptions have five increasing levels of intensity. Figure 2 shows the effects of the 16 corruptions in CIFAR-10-C at the first, third, and fifth levels of intensity.

Given α∈(0,1) and predicted probabilities $p(y_{c}|x_{i})$ from a model for all K classes c∈{1,...,K}, the 1−α prediction set S for a sample $x_{i}$ is the minimum sized set of classes such that:(4)$$\sum_{c\in S}p\left(y_{c}|x_{i}\right)\geq 1-\alpha$$

This results in a set of size $k_{i}$, which consists of the largest probabilities in the full probability distribution over all classes $p(y_{c}|x_{i})$ such that 1−α probability has been accumulated. This inherently assumes that the labels are unordered categorical classes such that including classes 1 and *K* does not imply that all classes between are also included in the set S. Then we can define:**Coverage:** For each example in a dataset, we calculate the 1−α prediction set of the label probabilities, then coverage is what fraction of these prediction sets contain the true label.**Width:** The width of a prediction set is simply the number of labels in the set, |S|. We report the average width of prediction sets over a dataset in our figures.
Although both calibration [25] and coverage can involve a probability over a model’s output, calibration only considers the most likely label, and its corresponding probability, while coverage considers the top-$k_{i}$ probabilities. In the classification setting, coverage is more robust to label errors as it does not penalize models for putting probability on similar classes.

## 4. Results

### 4.1. Regression

Figure 4 plots the mean test set coverage and width for the regression methods we considered averaged over the nine regression datasets. Error bars demonstrate that for low-performing methods, such as ensembling, dropout, and LL dropout, there is high variability in coverage levels and widths across the datasets.

We observe that several methods perform well across the nine datasets. In particular, LL SVI, SVI, and GPs all exceed the 95% coverage threshold on average, and linear regression comes within the statistical sampling error of this threshold. Over the regression datasets, we considered, LL SVI had the lowest mean width while maintaining at least 95% coverage. For specific values of coverage and width for methods on a particular dataset, see Table A1 and Table A2 in Appendix A.

Figure 4 also demonstrates an important point that will persist through our results. Coverage and width are directly related. Although high coverage can and ideally does occur when width is low, we typically observe that high levels of coverage occur in conjunction with high levels of width.

### 4.2. MNIST

In the classification setting, we begin by calculating coverage and width for predictions from Ovadia et al. [14] on MNIST and shifted MNIST data. Ovadia et al. [14] used a LeNet architecture, and we refer to their manuscript for more details on their implementation.

Figure 5 shows how coverage and width co-vary as dataset shift increases. The elevated width for SVI on these dataset splits indicate that the posterior predictions of label probabilities were the most diffuse to begin with among all models. In Figure 5, all seven models have at least 95% coverage with a 15-degree rotation shift. Most models do not see an appreciable increase in the average width of the 95% prediction set, except for SVI. The average width for SVI jumps to over 2 at 15 degrees rotation. As the amount of shift increases, coverage decreases across all methods in a comparable way. In the rotation shifts, we observe that coverage increases and width decreases after about 120 degrees of shift. This is likely due to some of the natural symmetry of several digits (i.e., 0 and 8 look identical after 180 degrees of rotation).

SVI maintains higher levels of coverage but with a compensatory increase in width. In fact, there is a Pearson correlation of 0.9 between the width of the SVI prediction set and the distance from the maximum shift of 14 pixels. The maximum shift occurs when the original center of the image is broken across the edge as the image rolls to the right. Figure 3’s right-most example is a case of the maximum shift of 14 pixels on a MNIST digit. This strong correlation between width and severity of shift for some methods makes the width of a prediction set at a fixed α level a natural proxy to detect dataset shift. For this simple dataset, SVI outperforms other models with regards to coverage and width properties. It is the only model that has an average width that corresponds to the amount of shift observed and provides the highest level of average coverage.

### 4.3. CIFAR-10

Next, we consider a more complex image dataset, CIFAR-10. Ovadia et al. [14] trained 20-layer and 50-layer ResNets. Figure 6 shows how the width of the prediction sets increases as the translation shift increases. This shift “rolls” the image pixel by pixel such that the right-most column in the image becomes the left-most image. Temperature scaling and ensemble, in particular, have at least 95% coverage for every translation, although all methods have high levels of coverage on average (though not exceeding 95%). We find that this high coverage comes with increases in width as shift increases. Figure 6 shows that temperature scaling has the highest average width across all models and shifts. Ensembling has the lowest width for the methods that maintain coverage of at least 95% across all shifts.

All models have the same encouraging pattern of width increasing as shift increases up to 16 pixels, then decreasing. As CIFAR-10 images are 28 pixels in width and height, this maximum width occurs when the original center of the image is rolled over to and broken by the edge of the image. This likely breaks common features that the methods have learned for classification onto both sides of the image, resulting in decreased classification accuracy and higher levels of uncertainty.

Between the models which satisfy 95% coverage levels on all shifts, ensemble models have lower width than temperature scaling models. Under translation shifts on CIFAR-10, ensemble methods perform the best given their high coverage and lower width.

Additionally, we consider the coverage properties of models on 16 different corruptions of CIFAR-10 from Hendrycks and Gimpel [28]. Figure 7 shows coverage vs. width over varying levels of shift intensity. Models that have more dispersed points to the right have higher widths for the same level of coverage. An ideal model would have a cluster of points above the 95% coverage line and be far to the left portion of each facet. For models that have similar levels of coverage, the superior method will have points further to the left.

Figure 7 demonstrates that at the lowest shift intensity, ensemble models, dropout, temperature scaling, and SVI were able to generally provide high levels of coverage on most corruption types. However, as the intensity of the shift increases, coverage decreases. Ensembles and dropout models have, for at least half of their 80 model-corruption evaluations, at least 95% coverage up to the third intensity level. At higher levels of shift intensity, ensembles, dropout, and temperature scaling consistently have the highest levels of coverage. Although these higher-performing methods have similar levels of coverage, they have different widths.

We also present a way to quantify the relative strength of each method over a specific level of corruption. In Figure 8, for instance, we plot only the coverage and widths of methods at the third level of corruption and use the fraction of the points of a particular method that lie above the regression line. Methods that are more effective are providing higher coverage levels at lower widths and will have more points above this regression line.

For each of the five corruption levels, we calculated a regression line that modeled coverage as a function of width. Figure 9 presents the fraction of marginal coverages on various CIFAR-10-C datasets for each method that exceeded the linear regression prediction. The larger the fraction, the better the marginal coverage of a method given a prediction interval/set of a particular width. We observe that dropout and ensembles have a strong relative performance to the other methods across all five levels of shift.

Finally, we compared the relative rank order of these methods across coverage, as well as two common metrics in uncertainty quantification literature: Brier score and ECE. Figure 11 shows that the rankings are similar across methods. In particular, coverage has a nearly identical pattern to ECE, with changes only in the lower ranking methods.

### 4.4. ImageNet

Finally, we analyze coverage and width on ImageNet and ImageNet-C from Hendrycks and Gimpel [28]. Figure A1 shows similar coverage vs. width plots to Figure 7. We find that over the 16 different corruptions at 5 levels, ensembles, temperature scaling, and dropout models had consistently higher levels of coverage. Unsurprisingly, Figure A1 shows that these methods have correspondingly higher widths. Figure 10 reports the relative performance of each method across corruption levels. Ensembles had the highest fraction of marginal coverage on ImageNet-C datasets above the regression lines at each corruption level. Dropout, LL dropout, and temperature scaling all had similar performances, while LL SVI had a much lower fraction of marginal coverage above the regression lines. None of the methods have a commensurate increase in width to maintain the 95% coverage levels seen on in-distribution test data as dataset shift increases.

## 5. Discussion

We have provided the first comprehensive empirical study of the frequentist-style coverage properties of popular uncertainty quantification techniques for deep learning models. In regression tasks, LL SVI, SVI, and Gaussian processes all had high levels of coverage across nearly all benchmarks. LL SVI, in particular, had the lowest widths amongst methods with high coverage. SVI also had excellent coverage properties across most tasks with tighter intervals than GPs and linear regression. In contrast, the methods based on ensembles and Monte Carlo dropout had significantly worse coverage due to their overly confident and tight prediction intervals.

In the classification setting, all methods showed very high coverage in the i.i.d setting (i.e., no dataset shift), as coverage is reflective of top-1 accuracy in this scenario. On MNIST data, SVI had the best performance, maintaining high levels of coverage under slight dataset shift and scaling the width of its prediction intervals more appropriately as shift increased relative to other methods. On CIFAR-10 data and ImageNet, ensemble models were superior. They had the highest coverage relative to other methods, as demonstrated in Figure 9 and Figure 10.

An important consideration throughout this work is the choice of hyperparameters in most all of the analyzed methods makes a significant impact on the uncertainty estimates. We set hyperparameters and optimized model parameters according to community best practices in an attempt to reflect what a “real-world” machine learning practitioner might do: selecting hyperparameters based on minimizing validation loss over nested cross-validation. Our work is a measurement of the empirical coverage properties of these methods as one would typically utilize them, rather than an exploration of how pathological hyperparameters can skew uncertainty estimates to 0 or to infinity, while this is an inherent limitation in the applicability of our work to every context, our sensible choices will provide a relevant benchmark for models in practice.

Of particular note is that the width of a prediction interval or set typically correlated with the degree of dataset shift. For instance, when the translation shift is applied to MNIST, both prediction set width and dataset shift is maximized at around 14 pixels. There is a 0.9 Pearson correlation between width and shift. Width can serve as a soft proxy of dataset shift and potentially detect shift in real-world scenarios.

Simultaneously, the ranks of coverage, Brier score, and ECE are all generally consistent. However, coverage is arguably the most interpretable to downstream users of machine learning models. Clinicians, for instance, may not have the technical training to have an intuition about what specific values of Brier score or ECE mean in practice, while coverage and width are readily understandable. Manrai et al. [33] already demonstrated clinicians’ general lack of intuition about the positive predictive value, and these uncertainty quantification metrics are more difficult to internalize than PPV.

Moreover, proper scoring rules (e.g., Brier score and negative log-likelihood) can be misleading under model misspecification [34]. Negative log-likelihood, specifically, suffers from the potential impact of a few points with low probability. These points can contribute near-infinite terms to NLL that distort interpretation. In contrast, marginal coverage over a dataset is less sensitive to the impacts of outlying data.

In summary, we find that popular uncertainty quantification methods for deep learning models do not provide good coverage properties under moderate levels of dataset shift. Although the width of prediction regions do increase under increasing amounts of shift, these changes are not enough to maintain the levels of coverage seen on i.i.d data. We conclude that the methods we evaluated for uncertainty quantification are likely insufficient for use in high-stakes, real-world applications, where dataset shift is likely to occur. However, marginal coverage of a prediction interval or set is a natural and intuitive metric to quantify uncertainty. The width of a prediction interval/set is an additional tool that captures dataset shift and provides additional interpretable information to downstream users of machine learning models.

## Figures and Tables

**Figure 1 entropy-23-01608-f001:**
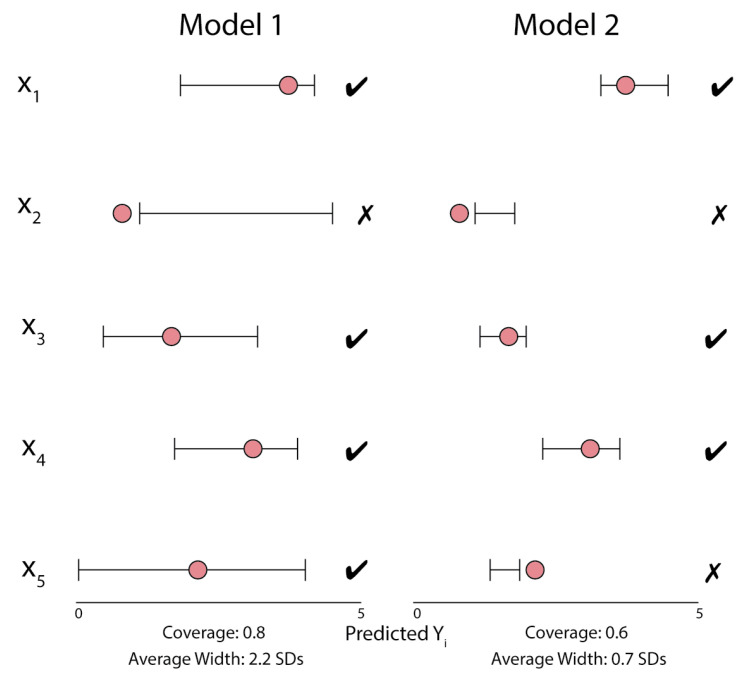
An example of the coverage properties for two methods of uncertainty quantification. In this scenario, each model produces an uncertainty interval for each $x_{i}$, which attempts to cover the true $y_{i}$, represented by the red points. Coverage is calculated as the fraction of true values contained in these regions, while the width of these regions is reported in terms of multiples of the standard deviation of the training set $y_{i}$ values.

**Figure 2 entropy-23-01608-f002:**
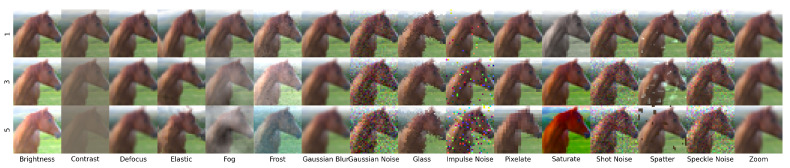
An example of the corruptions in CIFAR-10-C from [28]. The 16 different corruptions have 5 discrete levels of shift, of which 3 are shown here. The same corruptions were applied to ImageNet to form the ImageNet-C dataset.

**Figure 3 entropy-23-01608-f003:**
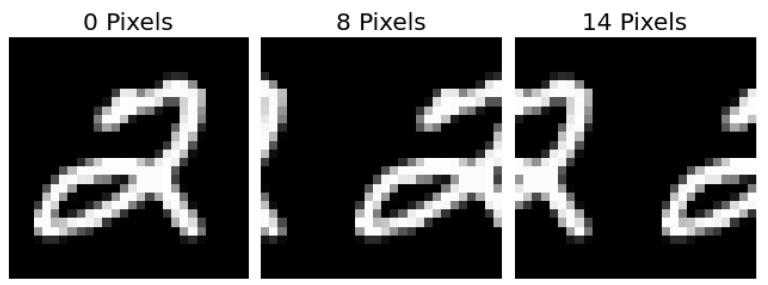
Several examples of the “rolling” translation shift that moves an image across an axis.

**Figure 4 entropy-23-01608-f004:**
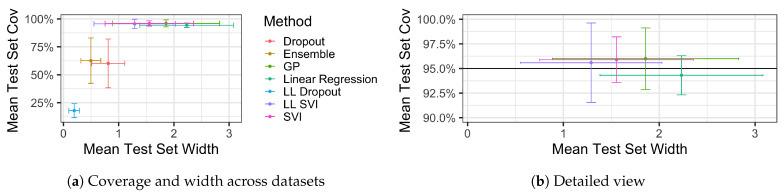
The mean coverage and widths of models’ prediction intervals average over the nine regression datasets we considered (panel **a**). Error bars indicate the standard deviation for both coverage and width across all experiments. In general, one would desire a model with the highest coverage above some threshold (here 95%) with a minimum average test set width. Models in the upper left have the best empirical coverage. In (panel **b**), we observe that the four methods which maintained 95% coverage did so because they had appropriately wide prediction intervals. LL SVI had the lowest average width while maintaining at least 95% coverage.

**Figure 5 entropy-23-01608-f005:**
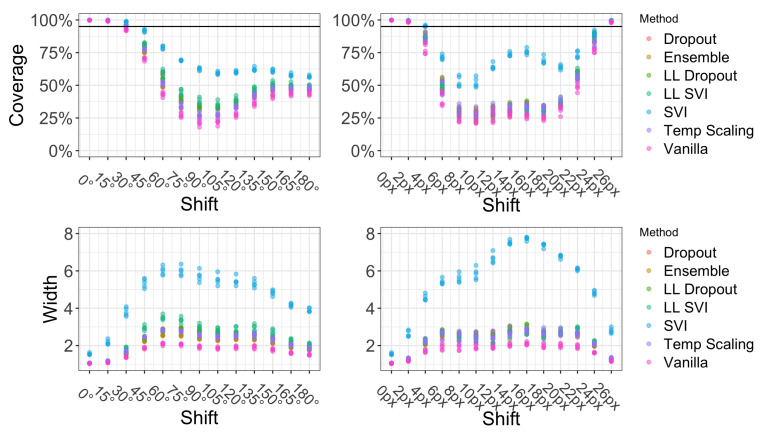
The effect of rotation and translation on coverage and width, respectively, for MNIST. 0 degrees or 0 pixels of shift indicate results on the test set of MNIST.

**Figure 6 entropy-23-01608-f006:**
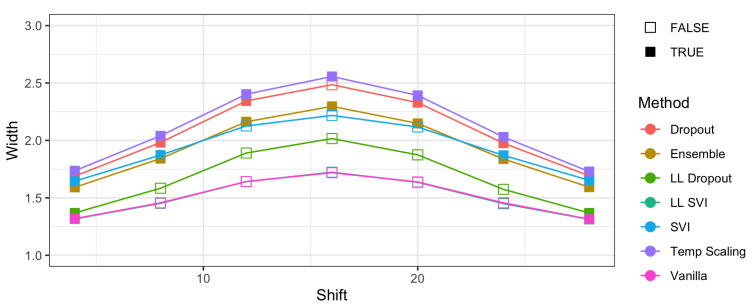
The effect of translation shifts on coverage and width in CIFAR-10 images. Coverage remains robust across all pixel shifts while width increases. The shading of points indicates whether 95% coverage was maintained when translated. In general, models with every point shaded maintain high levels of coverage. Therefore, the models with the best empirical coverage properties are the lowest width models such that coverage is maintained.

**Figure 7 entropy-23-01608-f007:**
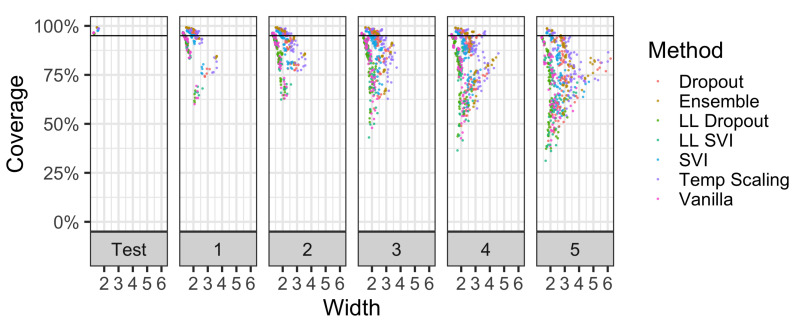
The effect of corruption intensity on coverage levels vs. width in CIFAR-10-C. Each facet panel represents a different corruption level, while points are the coverage of a model on one of 16 corruptions. Each facet has 80 points per method since 5 iterations were trained per method. For methods with points at the same coverage level, the superior method is to the left as it has a lower width. Please see Figure 8 and Figure 9 for the additional synthesis of these results.

**Figure 8 entropy-23-01608-f008:**
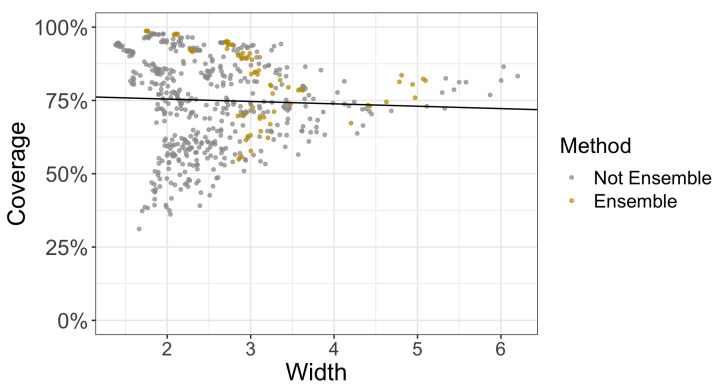
The coverage and width of ensemble and non-ensemble methods at the fifth level out of five levels of corruption in CIFAR-10-C. The black line is a simple linear regression of coverage against width. We then can consider the fraction of points for a particular method (in this case, ensembling) that are above the regression line (see Figure 9 and Figure 10). The higher the fraction of these points above the regression line, the better the method is at providing higher coverage at a relatively smaller width than other methods.

**Figure 9 entropy-23-01608-f009:**
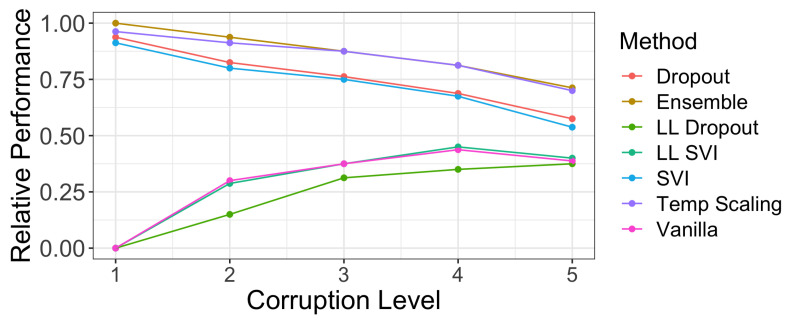
The fraction of marginal coverage levels achieved on CIFAR-10-C corruptions by our assessed methods that are above a regression line of coverage vs. width at a specific corruption level. Methods that have better coverage levels at the same width will have a higher fraction of points above the regression line (see Figure 8 for an example). At low levels of shift, dropout, ensemble, SVI, and temperature scaling have strictly better relative performance. As shift increases, poor coverage levels in general cause models to have more parity.

**Figure 10 entropy-23-01608-f010:**
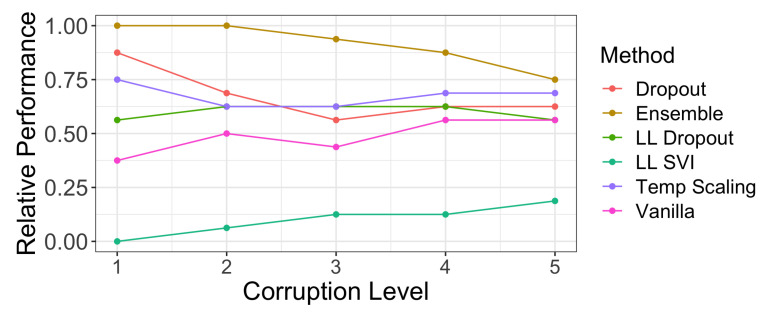
The fraction of marginal coverage levels achieved on ImageNet-C corruptions by our assessed methods that are above a regression line of coverage vs. width at a specific corruption level. Methods that have better coverage levels at the same width will have a higher fraction of points above the regression line (see Figure 8 for an example). Ensembling produces the best coverage levels given specific widths across all levels of corruption. However, at a higher level of dataset shift, there is more parity between methods.

**Figure 11 entropy-23-01608-f011:**
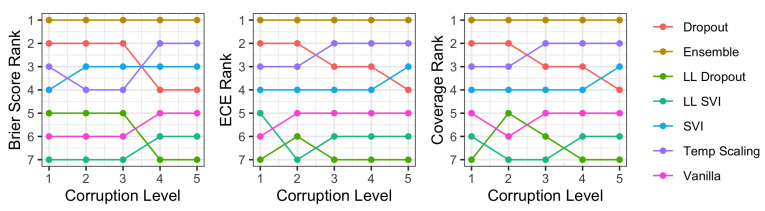
The ranks of each method’s performance with respect to each metric we consider on CIFAR-10-C. For Brier Score and ECE, lower is better, while for coverage, higher is better. We observe that all three metrics have a generally consistent ordering, with coverage closely corresponding to the rankings of ECE.

## Data Availability

Code is available at https://github.com/beamlab-hsph/coverage-quantification (accessed on 28 November 2021) with additional links to the public datasets, model parameters, and model predictions used in this work.

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
