# Peer review of "Empirical Frequentist Coverage of Deep Learning Uncertainty Quantification Procedures"

_entropy, 2021, doi:10.3390/e23121608_

Round 1
Reviewer 1 Report
The manuscript proposes to measure quality of uncertainty quantification (UQ) schemes for deep learning models using two criteria: coverage (of confidence sets/intervals) and their size. For the former, there is some desired level, such as 95%, and for the latter, smaller size is better, as long as the coverage is at least as large as the desired level. The authors use apply two criteria in several simulation studies to show that they shed light on the quality of various UQ approaches.
Overall, I think the two proposed criteria are attractive because of their simplicity and interpretability. Of course, they have been explored in other contexts, but using them as metrics for UQ is not common as of yet. The simulations presented in this work are interesting and well-done, and they convincingly show the value of the authors' proposed evaluation method.
A few major questions/concerns.
(1) This proposal is deeply connected to conformal prediction, but the literature review of that subject is nearly absent from this work. I'd recommend Vovk, Gammerman, and Shafer's "Algorithmic Learning in a Random World" as the most important, general-purpose citation. Another recent overview is "A Gentle Introduction to Conformal Prediction and Distribution-Free Uncertainty Quantification" by Bates and Angelopoulos. Various specific citations should be included elsewhere. For example, for line 116, see "Bayes-optimal prediction with frequentist coverage control" by Hoff. For a discussion of how conformal relates to distribution shift, highly relevant to the experiments presented herein, see "Robust Validation: Confident Predictions Even When Distributions Shift" by Cauchois, Gupta, Ali, and Duchi and "Conformal Prediction Under Covariate Shift" by Tibshirani, Barber, Candes, and Ramdas.
(2) The size metric is not directly comparable for models with different coverage levels. For this reason, size plots such as the one in Figure 5 are hard to interpret. To make the comparison of different methods more meaningful, why not calibrate them all to have the same coverage level (with conformal prediction) and then compare sizes?
Minor points
(1) At various points, the document says that it is desirable to have "high levels of coverage". E.g., Line 196 and the caption of Figure 4. This is a little imprecise. I think that one wants coverage as close as possible to the nominal level (e.g., 95%), not as large as possible. The wording should be adjusted in such places.
(2) Figure 7 is small and currently hard to read.
(3) In Figure 5, why not present the results with no shift as well? It would help provide context to report the coverage and size without any shift.
Reviewer 2 Report
Although this work proposes a study that seems novel, it is very complicated to understand it properly due to the way it is organised. For example, the introduction uses formulas that are unnecessary at the time and whose elements are described later. There is a Background subsection and then a Related Work section with a single subsubsection, which makes no sense. Point 3.1 concerns the experimentation carried out or the results section needs an introduction indicating how it is approached. In the same section it would be necessary to explain in detail the different datasets used. The benchmark used is taken from previous publications, however, due to the fact that these are conference publications and the fundamental role they play in the article, they should be explained in detail to show their suitability for the study.
Also noteworthy is the bibliography used, which makes excessive use of works published in conferences and needs to be updated to studies published in prestigious journals.
As a minor comment, the black line in figure 1.a makes it difficult to read.
Reviewer 3 Report
1. In the Relate Works, the references studies are in long paragraphs. It would be more comprehensive to organize a literature review with some pros and cons.
2. The main objectives and contributions are not apparent in the introduction part.
3. The regression tasks, LL SVI, SVI, and Gaussian processes all had high levels of coverage across nearly all benchmarks. Authors require to compare with other existing models with similar datasets.
4. How authors choose the parameter of regression and classification; discuss more detail about the parameters (Preprocessing).
5. Describe the fact for the large-scale evaluation of the empirical frequentist coverage properties of well-known uncertainty quantification techniques on a suite of regression and classification tasks.
6. The procedure our your approach should result systematically, and the functionality of your models should be clear to understand.
7. Their large-scale empirical study assessed these methods on standard datasets such as MNIST, CIFAR-10, ImageNet, and other non-image-based datasets. Additionally, how do these datasets' translations, rotations, and corruptions to quantify performance under dataset shift? Discuss the merit of these datasets' translations, rotations, and corruptions, etc.
Round 2
Reviewer 2 Report
The article has been suitably improved by addressing and responding to the different comments made.
Reviewer 3 Report
The author responded to the reviewer's comments very well. The article may accept in the current form.